# Direct imaging of delayed magneto-dynamic modes induced by surface acoustic waves

Michael Foerster[1], Ferran Macià [2,3], Nahuel Statuto[2,3], Simone Finizio[4,5], Alberto Hernández-Mínguez[6], Sergi Lendínez[3], Paulo V. Santos[6], Josep Fontcuberta[2], Joan Manel Hernàndez[3], Mathias Kläui [4] & Lucia Aballe[1]

The magnetoelastic effect—the change of magnetic properties caused by the elastic deformation of a magnetic material—has been proposed as an alternative approach to magnetic fields for the low-power control of magnetization states of nanoelements since it avoids charge currents, which entail ohmic losses. Here, we have studied the effect of dynamic strain accompanying a surface acoustic wave on magnetic nanostructures in thermal equilibrium. We have developed an experimental technique based on stroboscopic X-ray microscopy that provides a pathway to the quantitative study of strain waves and magnetization at the nanoscale. We have simultaneously imaged the evolution of both strain and magnetization dynamics of nanostructures at the picosecond time scale and found that magnetization modes have a delayed response to the strain modes, adjustable by the magnetic domain configuration. Our results provide fundamental insight into magnetoelastic coupling in nanostructures and have implications for the design of strain-controlled magnetostrictive nano-devices.

[1] ALBA Synchrotron Light Source, 08290 Cerdanyola del Valles, Spain. [2] Institut de Ciència de Materials de Barcelona (ICMAB-CSIC), Campus UAB, 08193 Bellaterra, Spain. [3] Dept. of Condensed Matter Physics, University of Barcelona, 08028 Barcelona, Spain. [4] Institut für Physik, Johannes Gutenberg Universität Mainz, 55099 Mainz, Germany. [5] Swiss Light Source, Paul Scherrer Institut, CH-5232 Villigen PSI, Switzerland. [6] Paul-Drude-Institut fur Festkörperelektronik, Hausvogteiplatz 5-7, 10117 Berlin, Germany. Michael Foerster and Ferran Macià contributed equally to this work. Correspondence and requests for materials should be addressed to F.M. (email: fmacia@icmab.es)

Magnetization states in magnetic materials are fundamental building blocks for constructing memory, computing, and further communication devices at the nanoscale. Static states such as magnetic domains are being used in non-volatile memories[1], whereas dynamic excitations—spin-waves—might serve to transmit signals and encode information in future electronic devices[2]. Collective magnetization states, which result from electron exchange coupling, are traditionally modified through magnetic fields created with electrical currents, giving rise to heat dissipation and stray fields. The spin-transfer-torque effect[3–5], which can be originated from pure spin currents, offers promising pathways toward the control of magnetic states at the nanoscale without using magnetic fields. Another promising strategy for handling high-speed magnetic moment variation at the nanoscale together with low-power dissipation is the use of electric fields. Although direct effects of electric fields on magnetic states are weak, electric fields can be used to induce strain and elastic deformations in a nanoscale magnetic material that might result in changes of magnetic properties as shown mostly by static experiments[6–12].

Surface acoustic waves (SAWs) are propagating strain waves that can be generated through oscillating electric fields at the surface of piezoelectric materials. SAWs have been used to induce magnetization oscillations in magnetic materials and to achieve assisted reversal of the magnetic moment[13–17]. However, SAW-induced magnetization dynamics is mostly treated as an effective variation in the magnetic energy, providing, thus, little information regarding the physical coupling between phononic and magnetization modes. In fact, delays between strain and magnetization dynamics were not considered. Resolving unequivocally the dynamic coupling between SAWs and magnetization requires resolving simultaneously strain and magnetization signals at the relevant time and space scales, which are picosecond and nanometer scales, respectively.

Here we report an experimental study providing a simultaneous direct observation of both strain waves and magnetization modes with high spatial and temporal resolution. Our technique combines time and spatially resolved photoemission electron microscopy (PEEM) with X-ray magnetic circular dichroism (XMCD)[18, 19] to achieve magnetic contrast. The low-energy secondary electrons detected by PEEM yield information on the piezoelectric potential caused by the strain wave in the piezoelectric substrate, thus providing a local measurement of strain strength. PEEM images taken with opposite X-ray photon helicities are combined to image the local magnetization through the dichroic effect (XMCD). Stroboscopic PEEM and XMCD images synchronized with the SAW allow us to correlate the local changes in magnetization with the spatial variation of the strain field.

## Results

**Imaging surface acoustic waves.** A schematic plot of the measurement is shown in Fig. 1. Micrometric Nickel (Ni) squares were deposited onto piezoelectric LiNbO$_3$ substrates containing interdigital transducers (IDTs) for the excitation of SAWs. The IDTs were designed to launch SAWs of a frequency $f_{SAW} = 499.654$ MHz at room temperature, which is exactly the repetition rate of X-ray bunches at the ALBA synchrotron in multibunch mode. By using an electronic phase locked loop between the synchrotron master clock and the rf-excitation signal applied to the IDT, we achieved phase synchronization between the SAW and the X-ray light pulses illuminating the sample[20] (see Methods section). For each phase delay between the SAW and the X-ray pulses, we recorded stroboscopic PEEM images that provided

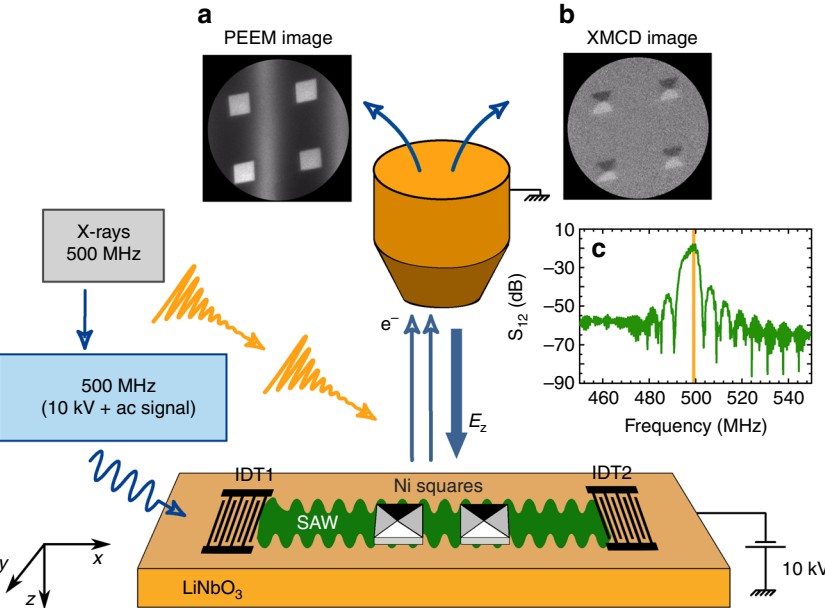

**Fig. 1** Schematic plot of the experimental setup. Circularly polarized X-rays illuminate the sample in the form of 20 ps pulses with a repetition rate of $f_0 \approx$ 500 MHz. The interdigital transducer, IDT1, of the hybrid device receives an AC electric signal of the same frequency, which is phase locked to the synchrotron repetition rate, generating a piezoelectric surface acoustic wave (*SAW*) that propagates through the LiNbO$_3$ substrate and interacts with the magnetic nanostructures. The phase-resolved variation of the piezoelectric voltage at the surface sample is probed with the PEEM, as well as the magnetization contrast along the X-ray propagation direction arising from the XMCD effect. In **a**, a PEEM image with a field of view of $20 \times 20$ μm$^2$ shows four $2 \times 2$ μm$^2$ Ni squares in presence of a piezoelectric wave front—*black* and *white stripes* indicate the sign of the piezoelectric voltage. In **b** an XMCD image of the same structures showing magnetic domains within the Ni squares. In **c** experimental rf-power spectrum of the transmission coefficient between IDT1 and IDT2, $S_{12}$, tuned to have a maximum at $f_0$. The resonance frequency $f_0$ is marked by a *yellow line*

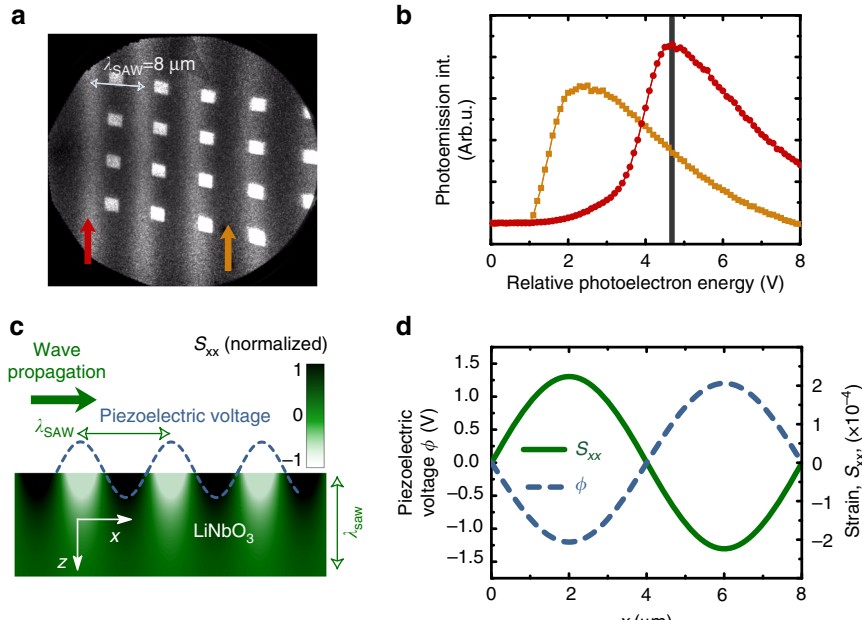

**Fig. 2** Images of SAW. **a** PEEM image of multiple Ni squares of $2 \times 2\ \mu m^2$ with a field of view of 50 μm recorded at a photoelectron energy of 4.8 V. The piezoelectric voltage produces periodic *dark* and *bright* zones in the PEEM image with the periodicity of the SAW, $\lambda_{SAW} = 8\mu m$. **b** Two photoelectron energy scans of the photoemission intensity corresponding to a *bright* and a *dark* zone (indicated in **a** with *red* and *orange arrows*). The *vertical line* indicates the electron energy at which **a** was acquired. **c** Schematic plot of the in-plane strain (in *green colorscale*) induced modulation in the LiNbO$_3$ caused by the acoustic wave. *Dashed blue line* indicates the oscillating piezoelectric voltage modulation. **d** Calculation of the in-plane strain component (*right hand side axis*) and the piezoelectric voltage (*left hand side axis*) at the sample surface for a SAW with a $\lambda_{SAW} = 8\ \mu m$ wavelength. Calculations of the strain are done to match the measured piezoelectric voltage of **a**. The magnetization changes are driven basically by the in-plane strain component along the SAW propagation direction. The piezoelectric voltage is in phase with the in-plane strain component, albeit with an opposite sign

magnetic contrast of the sample surface through the XMCD effect. Varying the SAW phase at constant power and after allowing for thermalization ensures that the magnetic changes observed are not due to thermal excitation. These stroboscopic measurements allow us to reconstruct the strain wave propagation and its effect on the magnetic structures with a time resolution of ≈80 ps.

In Fig. 2a we show a PEEM image with a field of view of 50 μm containing Ni squares of $2 \times 2\ \mu m^2$ in presence of SAWs. We observe bright and dark stripe lines with the periodicity of the SAW excitation (wavelength, $\lambda_{SAW} \approx 8\ \mu m$). The SAW produces a contrast in the PEEM images because the piezoelectric voltage associated with the wave shifts the energy of the secondary electrons that leave the sample surface. Imaging with a fixed phase delay and a slightly detuned (sub Hz) SAW frequency confirmed the SAW propagation direction by direct observation of the displacement of the stripes in consecutive PEEM images (Supplementary Movies 2). Figure 2b shows the number of secondary electrons (photoemission intensity) as a function of the electron kinetic energy, recorded by our detector at two surface areas corresponding to opposite phases of the wave, cf. the arrows in Fig. 2a. (Supplementary Movies 1). The energy shift between the two spectra corresponds to the peak-to-peak amplitude of the SAW-induced piezoelectric potential added to the 10 keV applied at the sample surface for PEEM detection (2.6 V for the rf-power used in Fig. 2a). A schematic plot of the oscillating piezoelectric SAW is presented in Fig. 2c showing the intensity of the strain modulation in the $x$–$z$ plane of a SAW propagating along $x$. The SAW decays exponentially with depth with a decay length of the order of the SAW wavelength[21, 22]. Figure 2c shows, as well, the oscillation

along $x$ of the piezoelectric potential at the sample surface (dashed blue line).

The measurement of the amplitude of the surface electric potential associated with the SAW allows for a quantification of the strain applied to the Ni nanostructures (Methods section). We plot in Fig. 2d the oscillation along $x$ of the piezoelectric potential with a peak-to-peak amplitude of 2.6 V (blue dashed curve) measured in Fig. 2a, together with its corresponding calculated longitudinal in-plane strain component, $S_{xx}$ (solid curve), which is the strain component responsible for the variations of the in-plane magnetic anisotropy in our structures. We notice that the piezoelectric potential, and therefore the out-of-plane electric field, $E_z = -\partial_z \phi_{SAW}$, is in phase with $S_{xx}$. Once we have shown that PEEM images provide a direct visualization of the surface potential associated to the SAW and thus a quantification of the dynamic strain, we now focus on the response originated by the SAW on the magnetic structures. The magnetoelastic effect (ME) arises from the coupling of spin moments to the lattice via orbital electrons. A change in the lattice caused by strain modifies bonds between magnetic atoms and changes the magnetic interactions, resulting in a ME-induced anisotropy.

**Imaging magneto-dynamic modes.** The intensity of the XMCD images is proportional to the component of the Ni magnetization along the X-ray incidence direction, represented in intensity gray scale. In order to quantify the ME-induced anisotropies, we chose polycrystalline Ni squares of $2 \times 2\ \mu m^2$ size and 20 nm thickness, having a four-domain Landau flux-closure state[12] (see images in Fig. 3). We first studied samples with the Ni squares sides aligned with the SAW propagation direction. Figure 3a, b shows a

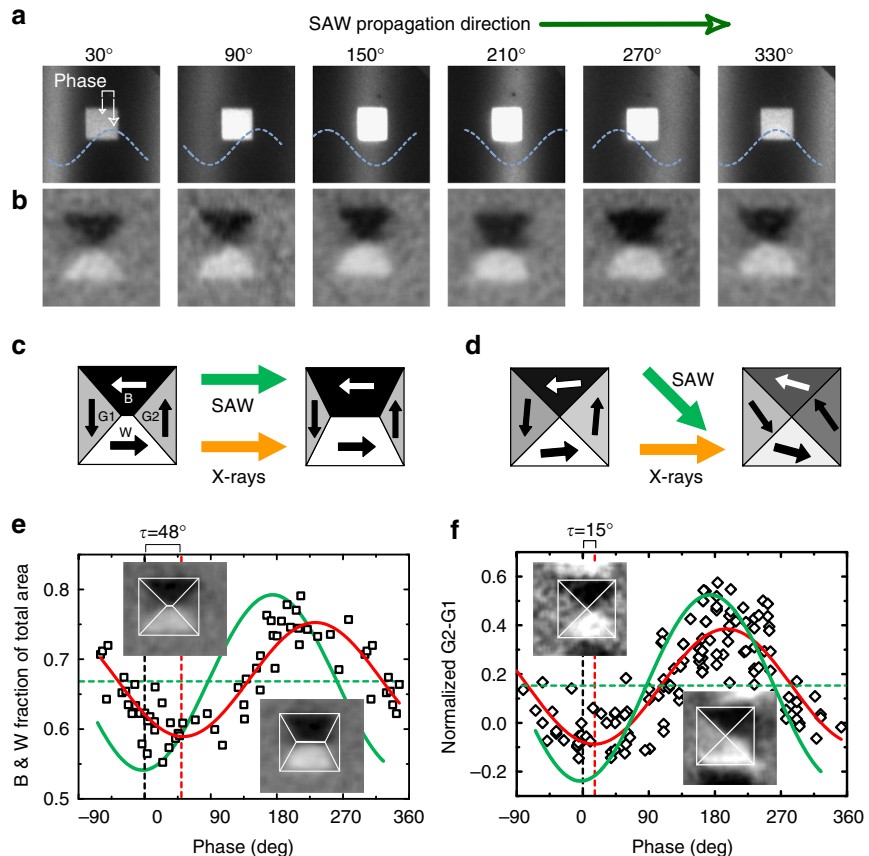

**Fig. 3** Simultaneous images of SAW and magnetic domains. **a** PEEM and **b** XMCD–PEEM images of a $2 \times 2\,\mu m^2$ Ni square at different phases of the SAW. Images correspond to phase lapses of 60° (that correspond to 333 ps). PEEM images are $8 \times 8\,\mu m^2$; XMCD–PEEM images are $4 \times 4\,\mu m^2$. **c** and **d** are schematic plots of the effect of SAW on Ni squares for configurations where SAW were aligned with the *squares' sides* **c** and with the *squares' diagonals* **d**. **e** Analysis of the domain configuration from multiple (4) Ni squares of $2 \times 2\,\mu m^2$ as a function of the individual phase with respect to the SAW for the configuration shown in **c**. We computed the area of *black* and *white* domains (amount of magnetization along the X-ray incidence direction). **f** Analysis of the domain configuration from multiple (9) Ni squares of $2 \times 2\,\mu m^2$ as a function of the individual phase with respect to the SAW for the configuration shown in **d**. We computed the intensity difference of the relative normalized *gray domains*, $(I(G1) - I(G2))/(I(W) - I(B))$. A best fit to the data with a sinusoidal function is plotted in *red* in both **b** and **c**. Delays of $\tau = 48° \pm 3$ and $\tau = 15° \pm 3$ with respect to the strain wave were obtained, respectively, for the two configurations. A schematic strain wave is plotted in *green* (with no scale) to indicate the phase values corresponding to maximum and minimum strain

temporal reconstruction of the effect of a SAW on a single Ni square; we plotted the direct PEEM images (Fig. 3a) to see the SAW propagation and the XMCD-PEEM images (Fig. 3b) to show the magnetic domain configuration (Supplementary Movie 3). The dynamical process of the magnetization is precisely observed in this figure where images correspond to intervals of 333 ps (1/6 of the SAW period): gray domains are first favored (magnetization perpendicular to SAW propagation) whereas black and white domains grow at larger values of the phase (magnetization aligned with the SAW propagation).

We have analyzed the response to SAWs of domain configurations from multiple squares within the same piezoelectric substrate by acquiring $20\,\mu m^2$ size XMCD images at different phase delays between SAW and X-ray pulses. At each phase delay, we calculated the area occupied by black and white domains in each square. This corresponds to the total area with magnetization oriented along the $x$ direction, which is the one modulated by the SAW. Figure 3e shows a summary of the obtained values as a function of the SAW phase acting on each square (measured from the direct PEEM image). The ensemble of points obtained from all analyzed squares is well fitted by a sinusoidal function (red curve) with the same periodicity as the

one of the SAW (green shadowed curve). We can translate the variations in the magnetic domain configuration observed in the XMCD images into variations of magnetic anisotropy by means of micromagnetic simulations (Supplementary Fig. 1). In the case shown in Fig. 3e, the oscillation of the domain areas is well reproduced by a strain-induced modulation of the magnetic anisotropy of amplitude $k_{ME,ac} \approx 1\,kJ\,m^{-3}$ superimposed to a preexisting uniaxial anisotropy of $k_U \approx 1.2\,kJ\,m^{-3}$ caused by the deposition process (shown as a dashed line at 0.67 in Fig. 3e). The static value of the preexisting in-plane uniaxial anisotropy has been confirmed by direct ferromagnetic resonance spectroscopy (FMR) measurements on the same films. We also estimated the in-plane strain at the surface of the substrate from the SAW piezoelectric potential as described in Fig. 2. For the experimental values reported in Fig. 3e, the corresponding value of the strain modulation is $S_{xx} = 4.5 \times 10^{-4}$.

From the correlation between in-space variations of strain and variation of magnetic anisotropy, we obtained a value for the parameter $\beta = k_{ME,ac}S_{xx}^{-1} = 2.2 \times 10^6\,Jm^{-3}$ at 500 MHz. The value of this ME coupling coefficient is similar to the reported values measured with static strain[12]. We expected the value $\beta$ to be similar to the static case because strain-induced changes of

magnetic anisotropy are related to the modifications of electron orbitals and thus these electronic properties must respond much faster than the 500 MHz strain oscillation used in our experiment. However, we observe in Fig. 3e (and also in Fig. 3a, b) a considerable delay between the magnetization oscillation and the strain wave that amounts to ≈270 ps (phase delay of ≈48°), which cannot be attributed to a delay in the sound propagation from the $LiNbO_3$ substrate to the Ni structures.

Phase delays can be expected if induced excitations (SAWs) are coupled to internal resonances of the system[23]. We have identified three different magnetic resonance processes upon a varying uniaxial anisotropy in the studied sample configuration with micromagnetic modeling (Supplementary Note 1), which correspond to precessional motion within the magnetic domain at ~2 GHz, domain-wall resonance at ~500 MHz, and vortex motion at ~50 MHz. Our micromagnetic model considers dynamic anisotropy variations in the Ni nanostructures at different frequencies and with a fixed spatial variation given by the wavelength, $\lambda \approx 8 \, \mu m$. Although there are models that consider changes in strain caused by changes in magnetization[24, 25], we estimated this effect to be small in our samples (Supplementary Note 2) and thus our model does not consider this second-order effect on the magnetization. The investigated system behaves similarly to a forced oscillator: at frequencies below resonance the magnetization configuration follows the changes in anisotropy induced by SAWs with no delay. At resonance the system follows the anisotropy changes with a delay (ideally 90°)—and with an increase of the oscillation amplitude that depends on the viscosity of the system (damping). At high frequencies the magnetization cannot follow the rapid changes in anisotropy and the oscillation amplitude of the magnetization vanishes and has large delays (approaching 180°). We note here that the domain-wall resonance has an excitation frequency close to the SAW excitation frequency, which can explain the measured delay.

In order to disentangle the different effects, we explored a judiciously selected different geometry consisting of Ni squares rotated 45° with respect to the SAW propagation direction in order to avoid the effect of domain-wall resonances (the samples had the same in-plane-induced anisotropy in the SAW propagation direction). Such a configuration has four magnetic domains energetically equivalent with respect to the in-plane uniaxial varying anisotropy induced by SAW and thus no domain growth (and shrinking) or domain-wall displacement occur; instead there is a coherent rotation of the magnetization within each of the domains. Our micromagnetic simulations show for this configuration that only precessional motion within the magnetic domain at ~2 GHz and vortex motion at ~50 MHz are present. We expect thus for such configuration a very small delay between the strain oscillation and the magnetization oscillation. The experimental study of this second configuration is plotted in Fig. 3f. To analyze the magnetic dynamics, the measurement of the black and white domain areas is not an adequate observable in this configuration because domains do not grow and shrink. Instead we plotted at each phase delay the normalized difference in intensity, $I$, of the two gray domains, (G1 and G2) normalized by the intensity of the overall black and white domains (B and W). This quantity is not zero in absence of strain (0.18 instead) due to the presence of the growth-induced anisotropy. The magnetic response is much faster in this configuration, showing a sizable decrease in the delay between the SAW and the magnetization oscillation, from 270 ps down to about 90 ps (phase delay of ≈16°), close to the experimental time resolution.

In brief, a macroscopic picture that explains intuitively the coupling between SAW and magnetization oscillations in the two presented configurations is the following. A varying uniaxial anisotropy creates a varying effective field that in the case of SAWs aligned with the squares' sides creates no torque to any of the four domains because either the field is parallel to the domain magnetization or it vanishes to zero. Instead, the effective field creates torques on the domain-wall magnetization, which causes a shift of the domain wall resulting in a magnetic domain growing or shrinking. In the second studied configuration where SAWs were aligned with the squares' diagonals, the varying anisotropy field induced by the SAWs form a 45° with domain magnetization, creating a torque that is equivalent in all four domains; and thus magnetization within domains rotates coherently and no domain-wall shifting occurs.

**Conclusion.** In summary, we have resolved simultaneously at the picosecond and nanometer scale the strain caused by a SAW of ≈500 MHz and the response of magnetic domains by using XMCD–PEEM microscopy, unveiling the dynamic response of the ME. We found that manipulation of magnetization states in ferromagnetic structures with SAWs is possible at the picosecond scale with efficiencies as high as for the static case. The magnetization dynamics are governed by the intrinsic configuration of the magnetic domains and by their orientation with respect to the SAW-induced strain resulting in considerable delays between strain and magnetization that must be considered in the design of magnetic devices. The described experiments offer an approach for the study of physical effects that depend on dynamic strain, as the concept may be applied to a wider range of research fields such as crystallography, nanoparticle manipulation, or chemical reactions[26, 27].

## Methods

**Sample fabrication and characterization.** Unidirectional IDTs were patterned with photolithography and deposited with electron beam evaporation (10 nm Ti| 40 nm Al| 10 nm Ti) on the 128° Y cut of $LiNbO_3$ piezoelectric substrates. The transducers generate multiple harmonics of a fundamental frequency, which was tuned such that the 4th harmonic matches the synchrotron frequency of 499.654 MHz. About 20 nm thick Ni nanostructures were defined with electron beam lithography and deposited by means of e-beam evaporation onto the piezoelectric substrate and in the acoustic path between the two IDTs (Fig. 1). The acoustic wave transmission from IDT1 to IDT2 was characterized both at atmospheric pressure with pico-probes and in the UHV chamber in the beamline with a network analyzer. The Ni films were characterized with SQUID magnetometry to determine the saturation magnetization, $M_s = 490 \times 10^3 \, A \, m^{-1}$ and with FMR spectroscopy to determine the damping parameter and the growth-induced in-plane uniaxial anisotropy, $\alpha = 0.03$ and $k_U = 1200 \, J \, m^{-3}$.

**Experimental realization.** The experiments were performed at the CIRCE beamline of the ALBA synchrotron light source[19]. The beamline employs an Elmitec spectroscopic low-energy electron and photoemission electron microscope (SpeLEEM/PEEM) operating in ultra-high vacuum. In order to generate XMCD-PEEM images, two PEEM images at the energy of the Ni $L_3$ absorption edge are acquired with opposite photon helicity (circular polarization), and then subtracted pixel-by-pixel to provide images with XMCD magnetic contrast as intensity. Samples were mounted on in-house designed sample holders[20] and the IDT contacted with wire-bonds. IDTs were several mm away from the sample center and the Ni nanostructures, thus allowing to screen them from the high electric field of the objective lens by the raised sample holder cap. After introduction in vacuum, each sample was degassed at low temperature (<100 °C) for at least 1 h in order to reduce the risk of arcs between sample (at high voltage) and microscope objective (at ground). A reduced acceleration voltage of 10 kV (standard is 20 kV) was used to further reduce the risk of discharges. The beamline intensity was adjusted in order to avoid excessive surface charging of the $LiNbO_3$ substrate. We used carbon doped $LiNbO_3$ substrates (known as black lithium niobate), mainly in order to reduce charging artifacts in PEEM imaging. We deposited thin metal lines connected to the sample holder potential close to the imaged area in order to reduce the charging effects and improve image resolution. For the synchronized excitation, the digital timing signal provided by the ALBA timing system[28] was converted into a phase locked 499.654 MHz (referred to as 500 MHz throughout the text) analog signal with a Keysight EXG Vector signal generator (model N5172B with option 1ER). The phase with respect to the master clock and the amplitude of the signal can be adjusted at this level as the experiment requires. The analog signal is then transmitted by a custom optical fiber system into the PEEM high-voltage rack and amplified[20]. The phase or temporal resolution

depends on the size of the zone analyzed as the phase changes with $\lambda$ approx 8 µm. However, an upper limit of the total temporal smearing (electronic jitter plus photon distribution from bunch length and bunch dephasing) for a small enough zone can be derived from the sharpest possible step feature of the SAW that can be resolved, which corresponds to ca. 80 ps. This is a good value for time resolved PEEM, which is helped by the continuous electrical AC excitation of a resonator structure (IDT), for which jitter is more easily controlled, but also reflects the quality of the ALBA beam in multibunch mode. All data presented were taken in thermal equilibrium. When the SAW were switched on, a slow (time scale of minutes) drift in the PEEM image was observed indicating a change of the $LiNbO_3$ surface temperature, affecting the SAW wavelength. Data were taken after a long period of thermalization and we compared snapshots at different instants within the 2 ns SAW cycle.

**Piezoelectric voltage to strain conversion**. We calculated the amplitude of the piezoelectric potential and strain tensor components by numerically solving the coupled differential equations of the mechanical and electric displacement for an acoustic wave propagating along the $x$ direction of a semi-infinite 128° Y-cut $LiNbO_3$ substrate (we used the values of density, dielectric constant, elastic, and piezoelectric coefficients from Roditi International Corporation). To obtain surface modes, we looked for solutions that decay toward $z > 0$ and satisfy the stress and electric displacement boundary conditions at the surface, $z = 0$. We have used the same SAW wavelength as the experiment, and have selected power density so that the amplitude of the simulated piezoelectric potential at $z = 0$ coincides with the measured one.

**Micromagnetic simulations**. Numerical simulations were performed using a MuMax3 code[29] on a graphics card with 2048 processing cores. A full code is appended in the Supplementary Note 3. We considered a two-dimensional layer and integrated the Landau–Lifshitz–Gilbert–Slonczewski equation to describe the magnetization dynamics. We computed different sample geometries with a resolution of 4 nm in the grid size. Thermal effects were neglected. The parameters of the magnetic layer, a Nickel film, were taken from the sample characterization: saturation magnetization $M_s = 490 \times 10^3$ A m$^{-1}$, Gilbert damping constant $\alpha = 0.03$, and exchange constant $A = 5 \times 10^{-12}$ J m$^{-1}$. Micromagnetic simulations with different values of $A$ (exchange constant), $M_s$ (saturation magnetization), and $\alpha$ (damping parameter) show little differences in the overall behavior: increasing $\alpha$ increases the width of resonances, increasing $M_s$ shifts the domain resonances (DR) toward higher frequencies and barely moves the domain-wall resonances (DWR), and increasing $A$ shifts the DWR toward higher frequencies and barely changes DR. We also note here that frequencies below the resonance produce a considerable change in the magnetization state but with phase shift (delay) that increases as we enter the resonance. A fixed uniaxial anisotropy is introduced in the simulations with a value $k_U = 1200$ J m$^{-3}$ with an additional oscillating term of $k_{ME,ac}$ having the wavelength set by the SAW ($\lambda_{SAW} \approx 8$ µm); simulations covered frequencies from 10s of MHz to 10s of GHz.

**Code availability**. The MuMax3 input file used for this study is given in Supplementary Note 3.

**Data availability**. The authors declare that data supporting the findings of this study are available in Supplementary Information files (Supplementary Movies 1–3). Additional data supporting the findings of the study and data on sample characterization are available from the corresponding author upon request.

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

## Acknowledgements

We thank Jordi Prat for technical help on the beamline and with the data analysis, Hermann Stoll and Rolf Heidemann for advice on electronics, Abel Fontsere, Bernat Molas, and Oscar Matilla from Alba electronics for the development of the 500 MHz synchronous excitation setup, and Werner Seidel from PDI for assistance in the preparation of the acoustic delay lines on LiNbO3. The project was supported by the ALBA in-house research program through IH2015PEEM and the allocation of in-house beamtime as well as with proposal 2016021647. F.M. acknowledges financial support from the Ramón y Cajal program through RYC-2014-16515. F.M. and J.F. acknowledge support from MINECO through the Severo Ochoa Program for Centers of Excellence in R&D (SEV-2015-0496). Funding through MAT2015-69144-P (J.M.H., N.S., and F.M.), MAT2015-64110-C2-2-P (L.A and M.F.) (MINECO/FEDER-UE) is acknowledged, and MAT2014-56063-C2-1-R (J.F.). J.F. also acknowledges support from the Catalan Governement through grant: 2014 SGR 734. S.F. and M.K. acknowledge Graduate School of Excellence Materials Science in Mainz (Grant No. GSC 266), the Swiss National Science Foundation (SNF), The German Research Foundation DFG (TRR 173 Spin+X), the ERC (ERC-2014-PoC 665672 MULTIREV), The EC (NMP3-LA-2010 246102 IFOX, FP-PEOPLE-2013-ITN 608003 WALL), and the Center for innovative and Emerging Materials at the Johannes Gutenberg Universität Mainz.

## Author contributions

M.F. and F.M. have contributed equally. M.F. conceived the PEEM experiment with input from S.F., J.F. and M.K. M.F., F.M., and L.A. planned and directed the project. L.A., M.F., F.M., N.S., S.L., J.M.H., and S.F. performed XPEEM measurements. L.A., M.F., F.M., and S.F. analyzed the data. N.S., J.M.H., and F.M. performed micromagnetic

simulations. A.H.-H. and P.S. designed, prepared, and characterized frequency tuned IDT on LiNbO₃, S.F. prepared the Ni microstructures. All authors discussed the results and contributed to drafting the manuscript.

## Additional information

**Competing interests:** The authors declare no competing financial interests.

