## [Peer Review File · Nature Communications]

Reviewers' comments:

Reviewer #1 (Remarks to the Author):

I am pleased to provide the following report of the manuscript "Direct Imaging of Delayed Magneto Dynamic Modes Induced by Strain Waves" by Foerster et.al. The manuscript describes a setup utilizing a time-resolved photoelectron emission microscopy to image the propagation of elastic waves along the surface of a piezoelectric material while simultaneously imaging the domain structure of an intimately coupled magnetic micro-square. The manuscript is well written and presents an interesting experimental observation utilizing a novel experimental setup. In general, I am positively inclined towards the technical aspects presented, but less so towards the scientific outcomes, which could be motivated and explained better.

Perhaps the most interesting portion of the paper is the phase lag between magnetization effects and elastic actuation. However, this experimental observation pops up in the last paragraph of page 6 with little to no further discussion of the effect. Based on the title of the manuscript, surely this is where the interest lies, and yet it is not well supported.

Three scenarios are presented to explain the effect, but a clear identification of the effect seems missing or timidly suggested without making a definitive assessment. For the case when SAW is parallel to the edge of the square, I think the explanation is that of domain wall motion, which from the calculation in the supplementary materials suggests that you may be able to catch the wings of the resonance at 500MHz. However, when the squares are rotated 45deg the domain wall resonance is suppressed and the domain precessional frequency (in - domain FMR?) shifts to ~ 3.5 GHz. Here there is nearly no overlap between the driving frequency (500MHz) and the resonance, so the efficiency of the process should reduce drastically, if present at all. Is it possible that this high frequency resonance just follows adiabatically the low frequency anisotropy and perhaps this is why the phase delay reduces? In this case it would not be resonant.

Furthermore, modeling the effect of the strain as a uniaxial anisotropy is interesting, but not presented in a physically intuitive way. I suspect you are referring to magnetoelastic energy considerations and effective fields that arise from them, in which case I agree to the direction of the strain relative to the wavevector. However, for a general audience this is not well supported or referenced.

Additional comments that may add to clarity:

1. I miss how the PEEM images for the elastic wave are acquired. I suspect that opposite helicity of the XMCD measurements at Ni L edge are added to give the PEEM image? This is not included in the text or methods.
2. By multibunch mode, I suspect you mean that the isolated pulse (cam bunch) is used and the PEEM is electronically gated to acquire only photoelectrons from that pulse, or does one gate on one of the multibunch buckets?
3. I am surprised that you can use an insulator in the PEEM without discharging. Were there any additional layers to protect against this?
4. There is a citation #18 referencing SAW. While ref 18 utilizes magneto elastics, it does not utilize it in the SAW configuration, but rather longitudinal. Hopefully, Scherbakov et.al. can forgive me if the citation is removed.
5. What accounts for the brightness or darkness of the PEEM images in 3A? For the 30deg and 330deg, the squares are dark as is the surrounding LiNbO₃, however, for 270deg, the surroundings are mostly dark, but the square is still bright. Is there processing done in these images or should one take their brightness at face value?
6. For the extracted value for anisotropy, the uniaxial is an out-of-plane uniaxial, while the ME anisotropy is an in-plane uniaxial? I miss the connection between the strain and anisotropy and why it is parallel to the wavevector (same as above)?
7. Is there an applied magnetic field?

8. On terminology, the Raabe paper refers to your mode as domain wall resonance, not precession. Precession elicits notions of something going around something else, which is not what is described in the reference (or here). Throughout the text, domain wall resonance and domain wall precession are used for the same effect. Also in the supplemental.

9. There is a statement in the introductory section "Moreover, studies on strain modulation of magnetic properties using SAWs struggled to disentangle thermal effects from elastic effects (some citations)...." I disagree with this statement. In most of those citations, thermal effects aren't discussed, so I don't see the relevance of this sentence. In fact, I would suggest that those citations explicitly show magnetoelastic coupling with no indication that the effects are thermally driven.

10. In figure 3, the phase plots represent different quantities. In one case it displays fraction of black and white and in the other grey – grey. For the sake of uniformity they should be the same or at least it should be explained why it is important to look at these different values for the two cases.

11. In figure 3, for the case of SAW || edge, it's not clear why the mean value is $\sim .67$ for the fraction of black and white domains, why not $.5$?

12. Similarly, for the case of SAW at 45deg to the edge, why is $G2 - G1$ not centered at 0?

In summary, I believe that there is something interesting in here, however, figure 3, which contains the interesting content, is not well supported or displayed without proper discussion. I think if some additional effort was devoted to describing the effects displayed in figure 3, then the manuscript would be strengthened tremendously.

Reviewer #2 (Remarks to the Author):

This paper reports the change of magnetization in ferromagnetic film induced by dynamic stress transfer. The authors use a piezoelectric crystal of LiNbO₃ containing interdigital electrodes for excitation of a conventional SAW structure. The produced surface acoustic waves are used to induce a change in the state of magnetization of the nickel squares placed inside the active part of the device. The original part of the paper is the use of time resolved PEEM and XMCD techniques: the former used to probe the variation of the piezoelectric voltage the latter for probing the magnetic contrast.

First of all, I appreciate to authors to their efforts on these experiments. They present a highly valuable set of data based on very rigorous novel experiments. Videos and supplementary materials provide useful complement of informations. Some comments arise from the document:

1- The surprising result comes from the observation between the two orientations of the Nickel squares (having a classical Landau domain structure). While the square having edges aligned with the SAW present a response delay of 270 ps, the square having the diagonal aligned with the SAW exhibit a reduced delay of 90ps. The experiments clearly show the contribution of the domain wall in the first case. In the second case, if coherent rotation is the only involved mechanism how the authors interpret the origin of the delay; it seems the rule out the damping of the magnetization in the Methods-Micromagnetic simulations section.

2- Strain modulation given in the text is of $4,5 \times 10^{-4}$, it would be useful to have the same scale on figure 2.

In conclusion, in my opinion, the paper brings enough new results for publication in Nature Communications.

Reviewer #3 (Remarks to the Author):

This is a technically very interesting work, combining high resolution magnetic imaging with a stroboscopic measurement technique. It is intriguing that surface polarization and magnetization can be measured in the same measurement allowing the authors to measure an absolute phase.

The data itself is quite convincing and generally of high quality. I do not agree however with the interpretation of the phase delay as being due to a hidden resonance of the magnetic layer, which is a fairly central point in this paper, and I cannot recommend publication until this has been resolved. The data by itself and without a detailed physical model of the dynamics is not sufficiently novel to accept this paper without an explanation of the dynamics, since strain coupling has been shown before, as the authors correctly state in their introduction.

The magnetic layer has a restoring force, the demagnetizing field, and does not necessarily behave like a harmonic oscillator, especially in geometry 3B. Magnetic structures show harmonic features in a FMR geometry when field and magnetization are orthogonal, they show friction-like dynamics when fields are applied parallel to the magnetization. Here, demagnetizing field (in-plane) and magnetization of two Landau domains (in-plane) are parallel and the result could be a friction-dominated dynamics, more like pulling a cart through mud than an oscillatory response, especially at these still relatively low frequencies. This reviewer is convinced that a connection between resonant damping and a phase shift has not been sufficiently demonstrated.

There is an obvious but perhaps not technically simple way of testing a resonance effect by increasing the SAW frequency until resonance occurs, leading to a large amplification of the magnetic excitation and 90 degree phase delay.

A more sophisticated model that includes strain and magnetic forces on the same footing could also help explain the dynamics, e.g: H. Sohn et al., ACS Nano 9, (2015)

We would like to thank the three referees for reviewing our manuscript and providing useful feedback, most of all by highlighting points where the clarity of our draft can be improved. In summary all three referees are very positive about our experiment and the quality and significance of the data obtained. However, while Referee 2 recommends the manuscript for publication, Referee 1 and 3 believe the interpretation and discussion of the results need to be improved. We have made revisions to the manuscript to address reviewers' questions, comments, and suggestions. We believe the new version is clearly improved and addresses all the points raised by the three referees. Below, we outline the main changes and how we have addressed the reviewers' comments.

Reviewer #1

I am pleased to provide the following report of the manuscript "Direct Imaging of Delayed Magneto Dynamic Modes Induced by Strain Waves" by Foerster et.al. The manuscript describes a setup utilizing a time-resolved photoelectron emission microscopy to image the propagation of elastic waves along the surface of a piezoelectric material while simultaneously imaging the domain structure of an intimately coupled magnetic micro-square. The manuscript is well written and presents an interesting experimental observation utilizing a novel experimental setup. In general, I am positively inclined towards the technical aspects presented, but less so towards the scientific outcomes, which could be motivated and explained better.

We thank the referee for the positive review and for the detailed comments. The referee has raised some questions that helped us to identify weak points in the manuscript. We have thus tried to clarify these points in the resubmitted version.

Perhaps the most interesting portion of the paper is the phase lag between magnetization effects and elastic actuation. However, this experimental observation pops up in the last paragraph of page 6 with little to no further discussion of the effect. Based on the title of the manuscript, surely this is where the interest lies, and yet it is not well supported.

We agree with the reviewer that this point was not emphasized enough in the first version of the manuscript—it is also for us an important part of our study. The new version has a more detailed description and a discussion on the origin of the phase lag between strain and magnetization (changes are highlighted in red in the manuscript).

Three scenarios are presented to explain the effect, but a clear identification of the effect seems missing or timidly suggested without making a definitive assessment. For the case when SAW is parallel to the edge of the square, I think the explanation is that of domain wall motion, which from the calculation in the supplementary materials suggests that you may be able to catch the wings of the resonance at 500MHz. However, when the squares are rotated 45deg the domain wall resonance is suppressed and the domain precessional frequency (in - domain FMR?) shifts to ~3.5 GHz. Here there is nearly no overlap between the driving frequency (500MHz) and the resonance, so the efficiency of the process should reduce drastically, if present at all. Is it possible that this high frequency resonance just follows adiabatically the low frequency anisotropy and perhaps this is why the phase delay reduces? In this case it would not be resonant.

The reviewer is right that in the case where the squares are rotated 45 deg and the phase delay is very small, the magnetization oscillation is not at resonance—and this is the reason why the delay is so small. The studied system could be seen as a forced oscillator. At low frequencies the magnetization configuration follows the changes in anisotropy with no delay. At resonances the system follows the anisotropy changes with a delay (ideally 90 deg)—and there is an increase of the magnetization oscillation amplitude that depends on the viscosity of the system (damping). At high frequencies the magnetization cannot follow the rapid changes in anisotropy and there is small amplitude of the magnetization oscillation amplitude with very large delay (approaching 180 deg). In summary the referee is providing precisely the correct explanation.

In the revised manuscript we have expanded the discussion on the origin of the phase lag between SAW and magnetization and added graphs in the Supplementary material that show the evolution of the magnetization with amplitude (showing real and imaginary parts) and phase as a function of frequency (See figure below). The new figure includes graphs of the same quantities shown for the experimental results: *i*) relative black and white area for the configuration with SAW along square sides and *ii*) difference of intensity within the two gray domains for the SAW along square diagonals. We can see in the new figure how the phase increases with frequency for the two

configurations as the system goes through the resonance: for the configuration with SAW aligned with square sides the phase begins to increase with the domain wall resonance (see panel A in the figure below) whereas for the configuration of SAW aligned with square diagonals there is no domain-wall resonance and the phase begins to increase at a higher frequency). Figure below has markers at $f = 500$ MHz to show the phase value for the two configurations (30 deg for SAW along square sides and 3 deg for SAW along square diagonals). We note that resonance frequencies (and thus phase values) are dependent on the material parameters. We have performed extensive micromagnetic simulations with different values of A (exchange), M_s (saturation magnetization) and α (damping parameter) and found the following overall effects on the domain resonance (DR) and domain-wall resonance (DWR): *i*) increasing α increases the width of the resonances, *ii*) increasing M_s shifts the DR towards higher frequencies and barely moves the DWR and *iii*) increasing A shifts the DWR towards higher frequencies and barely changes DR. We also note here that frequencies below the resonance produce a considerable change in the magnetization state but with phase shift (delay) that increases as we enter the resonance.

New Figure in Supplementary Materials. Amplitude, in phase (real), and out of phase (imaginary) parts (and phase in the lower panels) of the magnetic response of the system of $2 \times 2 \mu\text{m}$ Nickel squares to an oscillating anisotropy as a function of the frequency. Two configurations are plotted: in A the varying anisotropy axis is along the square sides, in B the varying anisotropy axis is along the square diagonals. We plotted for each configuration the same quantities we analyzed for the experimental data: in A we plotted the variation (in %) of the fraction area of black and white domains and in B, the variation (in %) of the normalized difference between gray intensities between the two gray domains. The lower panels showing the phase have dashed lines (black) marking the frequency of 500 MHz, which corresponds to the experiments: we see that for configuration of panels A the phase shift corresponds to 30 deg and almost no delay (3 deg) is observed for configuration in panels B.

We also provide in here the graph of the energy that was originally included in the initial version but with the used parameters in the revised version (we decreased the exchange constant from 1×10^{12} to 5×10^{11}). We believe that this graph is less visual and understandable than the amplitude and phase—and thus we are not plotting it in the revised version.

Old Figure from Supplementary Materials. Energy of the magnetic domain configuration of a Ni square $2 \times 2 \mu\text{m}^2$ with 20 nanometer in thickness as a function of the SAW frequency perturbation. The anisotropy is modulated with a fixed wavelength of $8 \mu\text{m}$ to emulate the effect of the SAW. The energy at low frequency is subtracted to all frequencies as a reference.

Furthermore, modeling the effect of the strain as a uniaxial anisotropy is interesting, but not presented in a physically intuitive way. I suspect you are referring to magnetoelastic energy considerations and effective fields that arise from them, in which case I agree to the direction of the strain relative to the wavevector. However, for a general audience this is not well supported or referenced.

The reviewer is correct with the interpretation. We have expanded the description of the modelling.

Additional comments that may add to clarity:

1. I miss how the PEEM images for the elastic wave are acquired. I suspect that opposite helicity of the XMCD measurements at Ni L edge are added to give the PEEM image? This is not included in the text or methods.

The elastic waves are detected from the piezoelectric voltage associated to the elastic deformation, i.e., the local surface potential changes to which the PEEM is very sensitive. The waves (if synchronized) show up in the direct PEEM image independent of the photon energy or polarization. So while magnetic contrast XMCD images were obtained by subtracting pixel-by-pixel two images at the Ni- L_3 absorption edge taken with opposite helicity (then dividing by the sum), the pixel-by-pixel sum (or average) of the same two images was used to determine the SAW position for the magnetic image. These images are shown in the Fig. 3A (upper row for PEEM images and lower row for XMCD images). We have rewritten the description of how images were taken.

2. By multibunch mode, I suspect you mean that the isolated pulse (cam bunch) is used and the PEEM is electronically gated to acquire only photoelectrons from that pulse, or does one gate on one of the multibunch buckets?

During the measurements, the synchrotron had a simple multibunch filling pattern, which is normally used at ALBA synchrotron. Note that the time/phase resolution of the experiment comes from the synchronization of the SAW to the RF cavities of the storage ring at 500 MHz, so that any electron that is on an orbit in the storage ring will

produce x-rays that sees the sample in the same state. Thus the full storage ring filling (all buckets) can be used for the stroboscopic imaging at 500 MHz. We did not use gating of any of the signals.

3. I am surprised that you can use an insulator in the PEEM without discharging. Were there any additional layers to protect against this?

Discharges are always an issue in PEEM and we took a few steps to reduce such a risk. We commented on this in the Methods section, Experimental realization: IDTs and wire-bonded contacts were hidden behind a conducting cap while only the central part of the sample was used for imaging (see Foerster *et al.* Ultramicroscopy **171**, 63 (2016)). We varied the gap between the cap and the sample surface so on the one hand it reduces imaging artefacts and beam shadowing and on the other hand it avoids short circuits between the cap and the bond wires. The PEEM extractor voltage was set to 10 kV instead of the standard 20 kV while maintaining the same working distance, which produces a loss of image resolution. Before measuring, samples were carefully degassed in the UHV preparation chamber at 40-70 C. The x-ray beam intensity was reduced two orders of magnitude by detuning of the undulator with respect to the monochromator. Additionally we used carbon-doped LiNbO₃ substrates (“black lithium niobate”), mainly in order to reduce charging artifacts in PEEM imaging. In some samples we deposited thin metal lines connected to the sample holder potential close to the imaged area in order to improve the images. However, we still had a few discharges, which did not damage the IDT structures.

4. There is a citation #18 referencing SAW. While ref 18 utilizes magneto elastics, it does not utilize it in the SAW configuration, but rather longitudinal. Hopefully, Scherbakov et.al. can forgive me if the citation is removed.

We removed the mentioned citation.

5. What accounts for the brightness or darkness of the PEEM images in 3A? For the 30deg and 330deg, the squares are dark as is the surrounding LiNbO₃, however, for 270deg, the surroundings are mostly dark, but the square is still bright. Is there processing done in these images or should one take their brightness at face value?

The images in Fig 3A were not processed. The Ni squares intensity reflects also the local LiNbO₃ surface potential. However, the metallic structure represents an electric equipotential surface (although the electric charge might not be uniform inside). We thus do not expect to see variation within the Ni squares with the PEEM images (and we indeed measured much larger squares of several tens of microns and the signal was always constant on the metal). From Fig 3A we see that at 30 deg the square is dark and at 150 and 210 it is bright and at 90 deg it is less bright. It seems that the bright color in the squares is reaching saturation, which makes it hard to distinguish between the bright at 150 deg and 210 deg and the less bright at 90 deg. The video detuned.avi shows how the wave passes through the Ni squares and this effect is clearly seen.

6. For the extracted value for anisotropy, the uniaxial is an out-of-plane uniaxial, while the ME anisotropy is an in-plane uniaxial? I miss the connection between the strain and anisotropy and why it is parallel to the wavevector (same as above)?

The film has an induced in-plane uniaxial anisotropy (not out-of plane) from the deposition process, which can be understood as a constant “offset” to the anisotropy generated by the ME. The ME creates an additional dynamic in-plane uniaxial anisotropy (and an out-of-plane as well but it is not relevant here) in the direction of propagation of the SAW with the same temporal character as the SAW. We modified the text to clarify this point in the new manuscript.

7. Is there an applied magnetic field?

In this experiment there was no applied magnetic field.

8. On terminology, the Raabe paper refers to your mode as domain wall resonance, not precession. Precession elicits notions of something going around something else, which is not what is described in the reference (or here). Throughout the text, domain wall resonance and domain wall precession are used for the same effect. Also in the supplemental.

We agree with the reviewer that the term precession is not appropriate for domain wall resonances. We have changed it.

9. There is a statement in the introductory section “Moreover, studies on strain modulation of magnetic properties using SAWs struggled to disentangle thermal effects from elastic effects (some citations)...” I disagree with this statement. In most of those citations, thermal effects aren’t discussed, so I don’t see the relevance of this sentence. In fact, I would suggest that those citations explicitly show magnetoelastic coupling with no indication that the effects are thermally driven.

We have modified the sentence to say that in our case we disentangled the thermal effects from the strain effects as we accessed to the effect of different phases of the wave (which correspond to different strain values), and removed the citations. However, without need to mention it, any experiment measuring the ME effect by SAW in an “on/off way” or as function of power (especially coercivity reductions) need to be very careful as the SAW definitely heat up the sample surface. Our experiment on the other side, compares data at constant SAW power (after thermalization) comparing different phases, and thus is intrinsically free of thermal effects.

10. In figure 3, the phase plots represent different quantities. In one case it displays fraction of black and white and in the other grey – grey. For the sake of uniformity they should be the same or at least it should be explained why it is important to look at these different values for the two cases.

In the first case the size of the domains (black, white, and gray) changes and the magnetization orientation within domains remains constant whereas in the second configuration the size of the domains does not change but the magnetization inside the domains slightly rotates (producing a variation in gray scale). We did not find an observable quantity that was equivalent in both configurations. We now emphasized in the revised manuscript why we chose such quantities in each case.

11. In figure 3, for the case of SAW || edge, it’s not clear why the mean value is ~.67 for the fraction of black and white domains, why not .5?

We probably did not make this point clear. There is an in-plane anisotropy of about 1000 J/m^3 arising from the deposition process which could be in any direction but happens to be close to the direction of propagation of SAW, and this is the reason that in Fig 3B the average value is not 0.5 and also in Fig 3C is not 0 (see also point 6). This value and direction has been confirmed by FMR measurement. In both cases, parallel and with 45 degree angle, the obtained mean value for either the black and white fraction or the normalized gray difference, correspond to the mentioned in-plane anisotropy value. We emphasized this in the revised version.

12. Similarly, for the case of SAW at 45deg to the edge, why is $G_2 - G_1$ not centered at 0?

See 11.

In summary, I believe that there is something interesting in here, however, figure 3, which contains the interesting content, is not well supported or displayed without proper discussion. I think if some additional effort was devoted to describing the effects displayed in figure 3, then the manuscript would be strengthened tremendously.

We have expanded the description of the model of varying anisotropies and introduced new micromagnetic data to clarify the origin of the delays.

Reviewer #2

This paper reports the change of magnetization in ferromagnetic film induced by dynamic stress transfer. The authors use a piezoelectric crystal of LiNbO₃ containing interdigital electrodes for excitation of a conventional SAW structure. The produced surface acoustic waves are used to induce a change in the state of magnetization of the nickel squares placed inside the active part of the device. The original part of the paper is the use of time resolved PEEM and XMCD techniques: the former used to probe the variation of the piezoelectric voltage the latter for probing the magnetic contrast.

First of all, I appreciate to authors to their efforts on these experiments. They present a highly valuable set of data based on very rigorous novel experiments. Videos and supplementary materials provide useful complement of informations.

We thank the referee for the review and for the positive comments. We are pleased that the reviewer considers our data valuable.

Some comments arise from the document:

1- The surprising result comes from the observation between the two orientations of the Nickel squares (having a classical Landau domain structure). While the square having edges aligned with the SAW present a response delay of 270 ps, the square having the diagonal aligned with the SAW exhibit a reduced delay of 90ps. The experiments clearly show the contribution of the domain wall in the first case. In the second case, if coherent rotation is the only involved mechanism how the authors interpret the origin of the delay; it seems the rule out the damping of the magnetization in the Methods-Micromagnetic simulations section.

Resonance processes occur in both configurations and delays of different amplitudes are thus expected in each configuration. See the responses to the first reviewer (on page 1 and 2) and the additional figure and discussion we included in the Supplementary materials, as well as in page 2 of this letter.

2- Strain modulation given in the text is of $4,5 \times 10^{-4}$, it would be useful to have the same scale on figure 2.

We have changed the scale in Fig 2D (so it is not in % now).

In conclusion, in my opinion, the paper brings enough new results for publication in Nature Communications.

Reviewer #3

This is a technically very interesting work, combining high resolution magnetic imaging with a stroboscopic measurement technique. It is intriguing that surface polarization and magnetization can be measured in the same measurement allowing the authors to measure an absolute phase.

The data itself is quite convincing and generally of high quality. I do not agree however with the interpretation of the phase delay as being due to a hidden resonance of the magnetic layer, which is a fairly central point in this paper, and I cannot recommend publication until this has been resolved. The data by itself and without a detailed physical model of the dynamics is not sufficiently novel to accept this paper without an explanation of the dynamics, since strain coupling has been shown before, as the authors correctly state in their introduction.

We thank the referee for reviewing our manuscript. We are pleased the referee considers our data convincing and of high quality.

The magnetic layer has a restoring force, the demagnetizing field, and does not necessarily behave like a harmonic oscillator, especially in geometry 3B. Magnetic structures show harmonic features in a FMR geometry when field and magnetization are orthogonal, they show friction-like dynamics when fields are applied parallel to the magnetization. Here, demagnetizing field (in-plane) and magnetization of two Landau domains (in-plane) are parallel and the result could be a friction-dominated dynamics, more like pulling a cart through mud than an oscillatory response, especially at these still relatively low frequencies. This reviewer is convinced that a connection between resonant damping and a phase shift has not been sufficiently demonstrated.

The reviewer is exposing here some ideas we have taken into account in the description of our data and modeling. However, we probably skipped some points in our arguments that made our explanation not easily comprehensible.

First of all, one should note that the restoring force of the demagnetizing field (and also anisotropy and exchange fields) is precisely what makes the system behave as an oscillator. Any small enough deviation from an equilibrium state produces a linear restoring force, making it a harmonic oscillator. Next, a friction-like phenomenon is included in the damping term of the oscillator. Now this damped oscillator is driven by an external periodic excitation, i.e. the SAW. The response of the oscillator will always be at the driving frequency, independent of its own internal resonances, but will change in phase and amplitude, as shown in the new figure of the supplemental information (see above in the response to Referee 1).

We agree that the effective fields arising from the change in anisotropy caused by SAW have only a torque effect in the magnetization of the domain walls for the configuration of Fig. 3B: the magnetization in the domain walls is tilted (but not perpendicular) with respect to the varying anisotropy axis (SAW propagation). Thus, as pointed out by the reviewer there is a clear torque on the domain wall magnetization (and indeed there is torque on all magnetization that is not collinear to the effective field). On the other hand the magnetization within domains is not affected by any torque from varying effective fields; in the case of the black and white domains because the magnetization and the anisotropy fields are collinear and in the gray domains because the anisotropy field vanishes (the anisotropy field is proportional to the magnetization in the direction of the varying anisotropy). In summary, in configuration Fig. 3B, the induced strain by SAW generates torques that act mainly on the domain-wall magnetization and the moving of domain walls causes a variation of the internal fields (both dipolar and exchange), which eventually cause a growth or shrink of the magnetic domains. In order to disentangle the different effects, we identified the configuration in Fig. 3C as a configuration where there is no torque in the domain-wall magnetization, in opposition to configuration 3B. In 3C there is an equivalent "direct" torque in all four domain magnetization, which combined with the damping tilts the magnetization towards the strain-induced easy axis.

Our modeling does consider the friction-like dynamics already (although for a proper friction dynamics simulation one might need to consider temperature and grain size distribution, which is something we believe it goes behind the scope of this study). For instance, at low frequencies the system follows the varying anisotropy field and at high frequencies the system does not follow it. We have added a new graph in the supplementary materials that shows how both amplitude and phase of the magnetic oscillation behave in response to the varying anisotropy as a function of the frequency. We thus emphasize that our model is a combination of friction-like dynamics and resonances (see response to reviewer 1).

Although we have first considered the possibility of a phenomenological friction-like dynamics $dm/dt = -\tau_0(m - m_{eq}(t))$, we finally decided that a micromagnetic model including the full LLG and thus resonances describes more accurately our experiment, since we are dealing with a fast periodic excitation (500 MHz) and not sharp pulses. We used micromagnetics because it is more precise to describe the magnetization dynamics than other more phenomenological approaches and it is a well-established modelling for this type of work. The simple

friction-like model does not explain the differences in the two configurations and would require a different viscosity for each one (we would have to assume that one viscosity corresponds to the domain walls and another to the magnetization inside each domain).

There is an obvious but perhaps not technically simple way of testing a resonance effect by increasing the SAW frequency until resonance occurs, leading to a large amplification of the magnetic excitation and 90 degree phase delay.

We agree that being able to change the frequency would provide further insight and we are working on it. However, it is a major task that will probably take one or two years, as bringing even higher frequency signals to the PEEM will require major hardware changes of the UHV system

A more sophisticated model that includes strain and magnetic forces on the same footing could also help explain the dynamics, e.g: H. Sohn et al., ACS Nano 9, (2015)

We agree with the reviewer that we should comment on the effect of magnetization on the strain wave as our model is considering only the evolution of magnetization under a varying anisotropy caused by the SAW. Although we are aware of the limitations of the presented model we consider that the effect of the magnetization changes on the strain are small and thus can be neglected in our system.

We have calculated the strain caused by the magnetization changes in our samples in both configurations following Ref. Nanotechnology 25 (2014) 435701. The strain is proportional to the magnetization with the magneto-mechanical coupling tensor λ (which we have taken from Nanotechnology 25 (2014) 435701 for Nickel). If we consider the strain variation caused in our sample between the two most different magnetization states (given by the two extremes of the SAW) we find that the maximum strain difference is $6e-5$ (for both S_{xx} and S_{yy} , not at the same time), which is about 13% of the strain we are applying ($4.5e-4$). Such a calculation would be valid for a sample with no boundary conditions but our sample is clamped to the LiNbO_3 surface and thus the real change of strain or deformation could be expected much lower. We thus consider that our model still describes accurately the presented experiment. We have added a discussion on this in the main manuscript with references to manuscripts where the model considering strain and magnetic forces is used (ACS Nano 9, 4814 (2015) and Nanotechnology 25 (2014) 435701). We also added in the supplementary material a calculation of the strain tensor and a figure that shows the strain differences induced in our samples due to the change in magnetic configurations.

New Figure in Supplementary Materials. Strain variation induced in a $2 \times 2 \mu\text{m}$ nickel square due to extremal changes in magnetization. The two magnetization states corresponding to the maximum and minimum values of anisotropy are plotted in the top panels in both configurations (a) for SAW propagation aligned with the squares side and (b) SAW propagation aligned with the squares diagonal. The bottom panel corresponds to the component S_{xx} of the strain tensor caused by the magnetization difference. It has a maximum value of $\pm 6e-5$ in the darkest/brightest areas.

REVIEWERS' COMMENTS:

Reviewer #1 (Remarks to the Author):

I provide here a referee report for the resubmitted manuscript "Direct imaging of delayed magneto dynamic modes induced by strain waves" by Foerster et.al. The manuscript describes experimental observations of coupled elastic and magnetic degrees of freedom in micrometer sized nickel patches.

Significant changes have been made to the manuscript since it's initial submission. I have two items that I disagree with on a conceptual level, but I don't think this warrants rejection of the work.

1. There are several places in the manuscript where notions of disentangling thermal effect are called upon. In my opinion, there is simply no basis for this. Everything here is in thermal equilibrium.

2. Somewhat more concerning is the statement in the introduction that says: "In fact, it is assumed implicitly that strain and induced magnetization dynamics occur instantaneously (i.e., without delays)." This is just incorrect. In fact, in any driven oscillator system phase delays are expected on either side of the resonance condition. Optically this phase delay can be measured with much higher precision than 80ps and can be seen easily.

Reviewer #2 (Remarks to the Author):

The paper in the present version is more easy to follow for non specialist. On the basis of the revised manuscript, the physical parameters involved in the coupling between the SAW filter and the nickel squares are described more satisfactorily.

The main contribution of this work is a clear demonstration of a new coupled technique between time resolved PEEM and XMCD allowing the dynamic investigation at nanoscale of the elastic coupling between ferroelectric and ferromagnetic materials.

The exposed work makes it possible to highlight the important magnetic parameters on the dynamics of the magneto-electric coupling in hybrid structures. The second aspect provides quantitative elements useful for dimensioning devices based on this coupling.

Few typos are present:

- page 7: " uniaxial varying anisotropy iduced by"
- in last sentence of the legend of Figure 2 of the supplementary material.
- also, since this figure gives a clear view of involved mechanisms, I suggest completing it with two inserts giving a schematic representation of the orientation of the nickel squares.

In conclusion, I feel that this paper brings enough novelty in the domain for publication in Nature Communications.

Reviewer #3 (Remarks to the Author):

The revised manuscript addresses my original concerns. The description of model and simulation are much improved. This is technically very interesting work and the paper is ready to be accepted in its current form. I would like to repeat that data for additional excitation frequencies is quite important to further understand the dynamics of the system and to clearly distinguish between resonant processes, off-resonance excited processes and strongly damped dynamics.

We would like to thank the referees for reviewing our manuscript again. We are pleased they found it suitable for publication in Nature Communications.

Reviewer #1

I provide here a referee report for the resubmitted manuscript "Direct imaging of delayed magneto dynamic modes induced by strain waves" by Foerster et.al. The manuscript describes experimental observations of coupled elastic and magnetic degrees of freedom in micrometer sized nickel patches.

Significant changes have been made to the manuscript since it's initial submission. I have two items that I disagree with on a conceptual level, but I don't think this warrants rejection of the work.

We thank the referee for reviewing again the manuscript.

1. There are several places in the manuscript where notions of disentangling thermal effect are called upon. In my opinion, there is simply no basis for this. Everything here is in thermal equilibrium.

The reviewer is right that in our experiment everything is in thermal equilibrium. We wanted to emphasize that in some previous studies (eg, APL Appl. Phys. Lett 88, 012503, 2006, which was coauthored by some of the authors of this manuscript) there was an effect of the temperature together with the effect of the strain. We have removed the sentence from the abstract and from the conclusion and just mentioned in the abstract that the experiments were done in thermal equilibrium.

2. Somewhat more concerning is the statement in the introduction that says: "In fact, it is assumed implicitly that strain and induced magnetization dynamics occur instantaneously (i.e., without delays)." This is just incorrect. In fact, in any driven oscillator system phase delays are expected on either either side of the resonance condition. Optically this phase delay can be measured with much higher precision than 80ps and can be seen easily.

We have modified the sentence and now it reads: "delays between strain and magnetization dynamics were not considered"

Reviewer #2

The paper in the present version is more easy to follow for non specialist. On the basis of the revised manuscript, the physical parameters involved in the coupling between the SAW filter and the nickel squares are described more satisfactorily.

The main contribution of this work is a clear demonstration of a new coupled technique between time resolved PEEM and XMCD allowing the dynamic investigation at nanoscale of the elastic coupling between ferroelectric and ferromagnetic materials. The exposed work makes it possible to highlight the important magnetic parameters on the dynamics of the magneto-electric coupling in hybrid structures. The second aspect provides quantitative elements useful for dimensioning devices based on this coupling.

We thank the referee for reviewing again the manuscript.

Few typos are present:

-page 7: "uniaxial varying anisotropy iduced by"

-in last sentence of the legend of Figure 2 of the supplementary material.

-also, since this figure gives a clear view of involved mechanisms, I suggest completing it with two inserts giving a schematic representation of the orientation of the nickel squares.

We have corrected the typos, the sentence in Fig S2 and introduced the schematic plots of the Ni squares.

In conclusion, I feel that this paper brings enough novelty in the domain for publication in Nature Communications.

Reviewer #3

The revised manuscript addresses my original concerns. The description of model and simulation are much improved. This is technically very interesting work and the paper is ready to be accepted in its current form. I would like to repeat that data for additional excitation frequencies is quite important to further understand the dynamics of the system and to clearly distinguish between resonant processes, off-resonance excited processes and strongly damped dynamics.

We thank the referee for the overall review.